# Positive Impact of Natural Deep Eutectic Solvents on the Biocatalytic Performance of 5-Hydroxymethyl-Furfural Oxidase

**Gonzalo de Gonzalo** [1],*[ID]**, Caterina Martin** [2][ID] **and Marco W. Fraaije** [2],*[ID]

[1] Departament of Organic Chemistry, University of Sevilla, c/ Profesor García González 1, 41012 Sevilla, Spain
[2] Groningen Biomolecular Sciences and Biotechnology Institute, University of Groningen, Nijenborgh 4, 9747AG Groningen, The Netherlands; c.martin@rug.nl
* Correspondence: gdegonzalo@us.es (G.d.G.); m.w.fraaije@rug.nl (M.W.F.);
  Tel.: +34-954-559997 (G.d.G.); +31-503-634345 (M.W.F.)

**Abstract:** Deep eutectic solvents (DESs) have been applied as cosolvents in various biocatalytic processes during recent years. However, their use in combination with redox enzymes has been limited. In this study, we have explored the beneficial effects of several DES as cosolvents on the performance of 5-hydroxymethylfurfural oxidase (HMFO), a valuable oxidative enzyme for the preparation of furan-2,5-dicarboxylic acid (FDCA), and other compounds, such as carbonyl compounds and carboxylic acids. The use of natural DESs, based on glucose and fructose, was found to have a positive effect. Higher conversions are obtained for the synthesis of several oxidized compounds, including FDCA. Depending on the type of DES, the stability of HMFO could be significantly improved. As the use of DES increases the solubility of many substrates while they only mildly affect dioxygen solubility, this study demonstrates that biocatalysis based on HMFO and other redox biocatalysts can benefit from a carefully selected DES.

**Keywords:** biocatalysis; oxidases; deep eutectic solvents; alcohol oxidation; green chemistry; hydroxymethylfurfural

## 1. Introduction

5-Hydroxymethylfurfural oxidase (HMFO, EC 1.1.3.47) from *Methylovorus* sp. strain MP688 is a flavin-containing oxidase that only requires molecular oxygen as a mild oxidant to catalyze various oxidation reactions which include selective oxidations of alcohols, aldehydes and even thiols [1–3]. Uniquely, this oxidative biocatalyst is capable of performing the three-step oxidation of 5-hydroxymethylfurfural (HMF), to the valuable furan-2,5-dicarboxylic acid (FDCA) [4], a chemical platform for the production of biobased polymers such as poly(ethylene-furandicarboxylate) (PEF). Several methodologies have been developed to obtain FDCA starting from HMF [5–7], with some examples including the application of enzymes [8–10], but with none of them was it possible to perform this synthesis effectively using a single biocatalyst. Thus, since HMFO is a valuable catalyst, the development of reaction conditions that can enhance its activity and/or stability are of high interest. To improve the efficiency of HMFO-based biocatalytic processes, HMFO mutants have been engineered to display increased (thermo)stability [11]. Recently, wild type HMFO and some of its mutants have been employed for the first time in the enantioselective oxidation of *sec*-allylic alcohols, achieving excellent enantioselectivities [12].

In order to develop effective HMFO-based processes, the use of non-conventional reaction media can be considered [13,14]. The application of such media can solve challenges related to limited

substrate and product solubility and biocatalyst activity and/or stability. In the last few years, deep eutectic solvents (DESs) have appeared as a promising alternative type of solvent [15–17]. DESs are typically prepared by the complexation of quaternary ammonium salts that act as hydrogen bond acceptors (HBAs) with hydrogen bond donors (HBDs). These solvents have similar properties to ionic liquids (ILs) [18,19], such as non-volatility, thermal stability and are liquids at temperatures lower than 100 °C. In addition, DESs can be prepared with high purity at low cost and present higher melting points when compared to ILs. In order to increase the number of DES and their applications, natural products such as amino acids, organic acids and urea or sugars have been employed for the preparation of DES, forming the so-called natural DESs (NADESs) [20,21]. These compounds show high diversity, are biodegradable and present very low toxicology, making them also attractive for pharmaceutical purposes. In the last few years, the applicability of (NA)DES has been demonstrated for organic extractions [22,23], for electrochemical purposes [24] and in catalytic procedures [25,26] Regarding this last application, DESs have been employed as solvents, cosolvents, additives or supports to anchor catalysts and/or solvents in a catalytic process.

Several examples have been described in which a (NA)DES was applied in the presence of different types of biocatalysts [27–30], including their use in redox biotransformations [31,32]. Specifically, NADESs have been employed in selective reductions catalyzed by alcohol dehydrogenases [33–35], as well as in oxidations using different types of enzymes [36–40]. Yet, until recently, no examples of the application of DESs as cosolvents in reactions catalyzed by flavin-containing oxidases have been reported. For this reason, we have analyzed the effect of different (NA)DESs in the oxidation of a variety of alcohols catalyzed by HMFO.

## 2. Results and Discussion

Initial experiments were focused on analyzing the substrate profile of HMFO when oxidizing a set of aromatic alcohols (alcohols with an aromatic moiety) at a concentration of 10 mM employing Tris/HCl buffer pH 8.0, containing 1% v/v DMSO, at 30 °C (Scheme 1). As can be observed in Table 1, benzyl alcohol (**1a**) was converted into benzaldehyde (**1b**) with high conversion (86% after 20 h). The presence of an electron-donating group at the aromatic ring led to a decrease in enzyme activity. Thus, after 24 h, only 42% conversion of 4-methoxybenzyl alcohol (**2a**) was observed. HMFO was also able to accept bulkier alcohols as substrates. For example, 4-phenylbenzyl alcohol (**3a**) was readily converted (77.2% conversion after 20 h). As can be observed in Entries 4 and 6, electron-withdrawing groups seem to have a positive effect on HMFO, as both 4-bromo- (**4a**) and 4-nitrobenzyl alcohol (**6a**) are oxidized with good conversions into the corresponding aldehydes (78.2% and 93.0% after 14 h, respectively). An exception to this behavior is 4-chlorobenzyl alcohol. This compound is oxidized to 4-chlorobenzaldehyde (**5b**) with only 31.7% conversion after 20 h. HMFO was able to convert cinnamyl alcohol (**7a**) into cinnamaldehyde (**7b**), as previously reported [2]. After 14 h, **7b** was recovered with 84% conversion (Entry 7, Table 1). No aldehyde formation was observed in the oxidation of the heteroaromatic alcohol 4-(4-pyridinyl)butan-1-ol (**8a**). The biocatalyst was also tested in the oxidation of two racemic primary alcohols, (±)-2-phenyl-1-propanol (**9a**), (±)-1-phenyl-2-propanol (**10a**) and the secondary alcohol (±)-1-phenylethanol (**11a**). For these three compounds, no activity was observed (data not shown in Table 1). In contrast, the benzofused secondary alcohol (±)-1-indanol (**12a**) was found to be a substrate for HMFO, though only resulting in a low conversion (7.0%) after 24 h (Entry 8, Table 1). For all the oxidations tested, no formation of the carboxylic acid from aldehyde oxidation was observed. This is in contrast with the conversion of other substrates of this oxidase. For example, HMF undergoes triple oxidation due to the formation of the hydrated forms of the formed aldehydes [1]. This suggests that the aromatic aldehydes formed in our experiments do not form appreciable amounts of the gem-diol form.

**Scheme 1.** Oxidation of aromatic alcohols **1-12a** catalyzed by HMFO in non-conventional media aqueous buffer; natural (NA)deep eutectic solvents (DESs).

**Table 1.** 5-hydroxymethylfurfural oxidase (HMFO)-biocatalyzed oxidations of aromatic alcohols (Tris/HCl 50 mM, pH 8.0, 30 °C) [1].

| Entry | Alcohol | Time (h) | Conv. (%) [2] |
|---|---|---|---|
| Entry 1 | **1a** | 20 | 86.3 ± 3.1 |
| Entry 2 | **2a** | 24 | 42.1 ± 1.6 |
| Entry 3 | **3a** | 20 | 77.5 ± 2.1 |
| Entry 4 | **4a** | 14 | 78.2 ± 1.3 |
| Entry 5 | **5a** | 20 | 31.7 ± 1.7 |
| Entry 6 | **6a** | 14 | 93.0 ± 1.4 |
| Entry 7 | **7a** | 14 | 85.5 ± 2.1 |
| Entry 8 | (±)-**12a** | 24 | 7.0 ± 1.3 |

[1] Reactions were carried out with substrate concentration 10 mM and 1.0 μM of HMFO in the presence of catalase from *Micrococcus lysodeikticus* (10 μL, 150,000 U/mL). [2] Measured by GC. Each reaction was performed at least twice.

(NA)DESs have emerged in recent years as a valuable reaction medium for biocatalyzed reactions in which also redox enzymes can be employed [31,32]. The presence of different (NA)DES in these reactions can present a beneficial effect on the activity and stability of the biocatalysts [41], allowing improved enantio- and diastereoselectivities [42] and even reversal on biocatalyst enantiopreference [35]. Regarding the application of oxidases, a laccase from *Bacillus* HR03, a multi-copper-dependent oxidase, has been recently tested in the presence of different (NA)DES. It was observed that this biocatalyst showed the highest activity in 20% *v/v* glycerol:betaine (2:1) [40]. This enzyme also presents higher stability in sorbitol:betaine:water (1:1:1) and glycerol:betaine (2:1) reaction media compared to aqueous buffer at 80 and 90 °C. In this study, the positive impact of the use of NADES in redox biocatalysis is shown for the first time for a flavin-containing oxidase.

In view of the results above, we decided to employ 4-chlorobenzyl alcohol (**5a**) as a model substrate to analyze the effect of DES on HMFO-based conversions, as aldehyde **5b** was obtained with a moderate conversion in the oxidation carried out in regular buffer conditions (Table 1, Entry 5). Reactions were carried out with a concentration of solvent equal or higher to 60% *v/v* to ensure a proper DES structure in the reaction medium [43]. All mixtures resulted in clear solutions with no indication of phase separation. Initial experiments (Table 2) were performed by using DES derived from choline chloride (ChCl) as hydrogen bond acceptor and different hydrogen bond donors, including polyols (glycerol (Gly) or ethyleneglycol (EG)), urea (U), acid (malonic acid (MA)) and sugars (D-xylitol (Xyl), D-sorbitol (Sor), D-glucose (Glu) or D-fructose (Fru)), at the corresponding molar proportions to form a DES. Oxidations performed with ChCl:Gly (1:2) or ChCl:EG (1:2) led to a slight increase in the enzymatic activity, as 4-chlorobenzaldehyde (**5b**) was recovered with 41%–45% conversion,

respectively (Entries 1 and 2, Table 2). No reaction was observed with the DES containing urea or malonic acid (Entries 3 and 4). The use of the sugar-based DES ChCl:Xy (1:1), led to a very low conversion (Entry 5). The substitution of xylitol by D-sorbitol has a positive effect on the activity, with a 54% conversion after 24 h (Entry 6). Choline chloride was also tested in mixtures with water and fructose or glucose (Entries 7 and 8, Table 2). The use of these DESs afforded excellent results, as in both cases 4-chlorobenzyl alcohol **5a** was completely oxidized to the corresponding aldehyde after 20 h. The use of a NADES not based on ChCl, was also tested. For this, a mixture containing D-glucose, D-fructose and water (Glu:Fru:H$_2$O) in a molar ratio of 1:1:6 was used. Again, the full conversion of **5a** was observed after 20 h (Entry 9, Table 2). Thus, the proper selection of a DES led to an important increase in HMFO performance. It is difficult to rationalize the beneficial effects of specific DESs on enzyme performance. It has been shown before that the use of DES can have a specific effect dependent on the enzyme studied [26]. In this case, it has been observed that HMFO can suffer from loss of the prosthetic group, the FAD cofactor [2]. It is tempting to conclude that some specific polyols (D-sorbitol) and carbohydrates (fructose and glucose) prevent conformations of the enzyme that allow dissociation of the cofactor, resulting in better performance.

**Table 2.** HMFO-biocatalyzed oxidation of alcohol **5a** in buffer/DES medium [1].

| Entry | DES | % DES (*v/v*) | T (°C) | Time (h) | Conv. (%) [2] |
|---|---|---|---|---|---|
| Entry 1 | ChCl:Gly (1:2) | 60 | 30 | 18 | 41.5 ± 2.1 |
| Entry 2 | ChCl:EG (1:2) | 60 | 30 | 18 | 45.0 ± 2.8 |
| Entry 3 | ChCl:U (1:2) | 60 | 30 | 24 | ≤3 |
| Entry 4 | ChCl:MA (1:1) | 60 | 30 | 24 | ≤3 |
| Entry 5 | ChCl:Xyl (1:1) | 60 | 30 | 24 | 8.0 ± 1.4 |
| Entry 6 | ChCl:Sor (1:1) | 60 | 30 | 24 | 54.5 ± 2.1 |
| Entry 7 | ChCl:Glu:H$_2$O (5:2:5) | 60 | 30 | 20 | ≥97 |
| Entry 8 | ChCl:Fru:H$_2$O (5:2:5) | 60 | 30 | 20 | ≥97 |
| Entry 9 | Glu:Fru:H$_2$O (1:1:6) | 60 | 30 | 20 | ≥97 |
| Entry 10 | ChCl:Gly (1:2) | 80 | 30 | 20 | 5.5 ± 0.7 |
| Entry 11 | ChCl:Gly (1:2) | 90 | 30 | 24 | ≤3 |
| Entry 12 | ChCl:Sor (1:1) | 80 | 30 | 24 | 12.0 ± 1.4 |
| Entry 13 | ChCl:Sor (1:1) | 90 | 30 | 24 | ≤3 |
| Entry 14 | Glu:Fru:H$_2$O (1:1:6) | 80 | 30 | 20 | 42.5 ± 2.1 |
| Entry 15 | Glu:Fru:H$_2$O (1:1:6) | 90 | 30 | 20 | ≤3 |
| Entry 16 | None | 60 | 60 | 8 | 55.0 ± 1.4 |
| Entry 17 | Glu:Fru:H$_2$O (1:1:6) | 60 | 60 | 8 | ≥97 |

ChCl: Choline chloride; Gly: Glycerol; EG: Ethylenglycol; U: Urea; MA: Malonic acid; Xyl: D-Xylitol; Sor: D-Sorbitol; Glu: D-Glucose; Fru: D-Fructose [1] Reactions were carried out with substrate concentration 10 mM and 1.0 µM of HMFO in the presence of catalase from *Micrococcus lysodeikticus* (10 µL, 150,000 U/mL). [2] Measured by GC. Average values of two or more experiments.

With the aim of determining the effect of the DES concentration on the biocatalytic oxidations, reactions were carried out with higher concentrations of ChCl:Gly (1:2), ChCl:Sor (1:1) and Fru:Glu:H$_2$O (1:1:6). For the glycine-containing DES, an 80% *v/v* concentration led to the poor performance of the biocatalyst, with only 5.5% of **5b** formed after 20 h, as shown in Entry 10 of Table 2. Low performance at 80% *v/v* was also observed for ChCl:Sor (1:1), resulting in 12.0% of aldehyde formed after 24 h. Regarding the glucose and fructose-based DES, the use of 80% *v/v* still led to a higher conversion when compared with buffer (42.5% conversion after 24 h, Entry 14, Table 2). For the three DESs analyzed, no oxidation was observed at 90% *v/v* concentrations. Thus, 80% *v/v* seemed to be the maximum amount of DES accepted by HMFO, which retains a good performance with Glu:Fru:H$_2$O (1:1:6).

Biooxidations employing HMFO in a mixture of Tris buffer (50 mM pH 8.0) and 60% *v/v* Fru:Glu:H$_2$O (1:1:6) were extended to the rest of the tested aromatic alcohols. As can be observed in Table 3, for all the substrates that were shown to be converted in aqueous medium, there was an increase in the conversions (see Table 1 vs. Table 3). Thus, the HMFO-catalyzed oxidation of benzyl alcohol (**1a**)

led to a complete conversion after 18 h. As this compound is more reactive than its chlorinated analog **5a**, it was possible to recover benzaldehyde (**1b**) with complete conversion even at 80% *v/v* of DES (Entries 1 and 2, Table 3). Higher concentrations of the DES led to a drop in the enzyme activity, but the biocatalyst is still active in 90% *v/v* DES with an observed conversion of 27.5% after 18 h (Entry 3, Table 3). 4-Methoxybenzaldehyde (**2b**) can be obtained with 78.3% conversion after 24 h, representing a great increase in the enzymatic activity when compared with the oxidation in buffer. In addition, the alcohols **3-4a** and **6-7a** showed excellent conversions after short reaction times, with conversions of around 90%. Thus, cinnamaldehyde (**7b**) was obtained at 96% after 14 h, as shown in Entry 9 of Table 3. Finally, as recently shown with a set of allylic secondary alcohols [12], HMFO was able to catalyze the kinetic resolution of the secondary racemic alcohol 1-indanol, (±)-**12a** (Entry 10, Table 3). The biocatalyst selectively oxidizes one of the enantiomers of the starting alcohol to 1-indanone (**12b**), whereas the remaining enantiomer of **12a** gets enantioenriched, as shown in Scheme 2. Thus, after treating racemic 1-indanol with HMFO for 24 h, 26% of 1-indanone was obtained. This represents an important increase in HMFO performance when compared with aqueous buffer as a medium where only 7% of 1-indanone was obtained (see Entry 8, Table 1). The optical purity of the remaining substrate at the end of the reaction was measured by HPLC, which revealed that (*R*)-1-indanol was recovered with 17% *ee*, showing a similar enantiopreference of HMFO when catalyzing the kinetic resolution of allylic alcohols [12]. With these values, it can be estimated that the enantioselectivity (*E*) [44] for this kinetic resolution was 21. This is a moderate value when compared with the previous report in which allylic secondary alcohols were studied as substrates [12]. This different behavior can be explained by the different structure of the starting alcohol, as 1-indanol is sterically constrained, complicating enantiodiscrimination by the biocatalyst. Still, the kinetic resolution of 1-indanol represents the first example of enantioselective oxidation of a benzofused racemic secondary alcohol catalyzed by HMFO [12]. With the available crystal structure of HMFO, enzyme engineering may be performed to improve the enantioselective behavior towards 1-indanol and related compounds [1].

**Scheme 2.** Kinetic resolution of racemic 1-indanol catalyzed by HMFO.

**Table 3.** HMFO-biocatalyzed oxidations of aromatic alcohols (Tris/HCl 50 mM, pH 8.0, 30 °C) using Glu:Fru:H$_2$O (1:1:6) [1].

| Entry | Substrate | % DES (*v/v*) | Time (h) | Conv. (%) [1] |
|-------|-----------|---------------|----------|---------------|
| Entry 1 | **1a** | 60 | 18 | ≥97 |
| Entry 2 | **1a** | 80 | 18 | ≥97 |
| Entry 3 | **1a** | 90 | 18 | 27.5 ± 0.7 |
| Entry 4 | **1a** | 95 | 24 | ≤3 |
| Entry 5 | **2a** | 60 | 24 | 78.3 ± 1.5 |
| Entry 6 | **3a** | 60 | 20 | 90.0 ± 1.4 |
| Entry 7 | **4a** | 60 | 14 | 92.6 ± 1.5 |
| Entry 8 | **6a** | 60 | 14 | 95.6 ± 2.1 |
| Entry 9 | **7a** | 60 | 14 | 95.6 ± 2.1 |
| Entry 10 | (±)-**12a** | 60 | 24 | 26.0 ± 1.7 |

[1] Determined by GC. Average value of two or more experiments.

The effect of the 4-chlorobenzyl alcohol concentration on the efficiency of conversion was analyzed when using merely buffer (50 mM Tris/HCl, pH 8.0) or DES in buffer (60% *v/v* of Fru:Glu:H$_2$O (1:1:6) or ChCl:Fru:H$_2$O (5:2:5)). In order to compare the results at different reaction times, the reaction rate has

been defined as the amount of mmoles of **5a** oxidized per liter and per hour (mmol/L h). As shown in Figure 1, the use of 60% *v/v* of both DESs at any substrate concentration led to higher rates when compared with the biooxidation in only buffer. When working with buffer alone, the highest rate was observed at a 10 mM substrate concentration (15 mmol/L h) and decreased down to 4.7 mmol/L h at 100 mM, while no conversion was observed at higher concentrations. A similar trend was observed for the oxidation in buffer with 60% *v/v* ChCl:Fru:$H_2O$ (5:2:5), with the highest reaction rate at 10 mM. However, in this medium all the reaction rates are higher than in only buffer, affording a reaction rate of 14.1 mmol/L h at 100 mM. The use of Fru:Glu:$H_2O$ (1:1:6) led to the highest reaction rates at all tested alcohol concentrations. In these conditions, the maximum value was obtained when working at 20 mM substrate (51.2 mmol/L h). Still, at 100 mM, a reaction rate of 23.4 mmol/L h was measured, a value that is still higher than the one obtained in the absence of DES with **5a** concentrations ten times lower. Even at 200 mM, the biocatalyst is able to oxidize the alcohol when using a DES, with the highest reaction rate of 10.4 mmol/L.

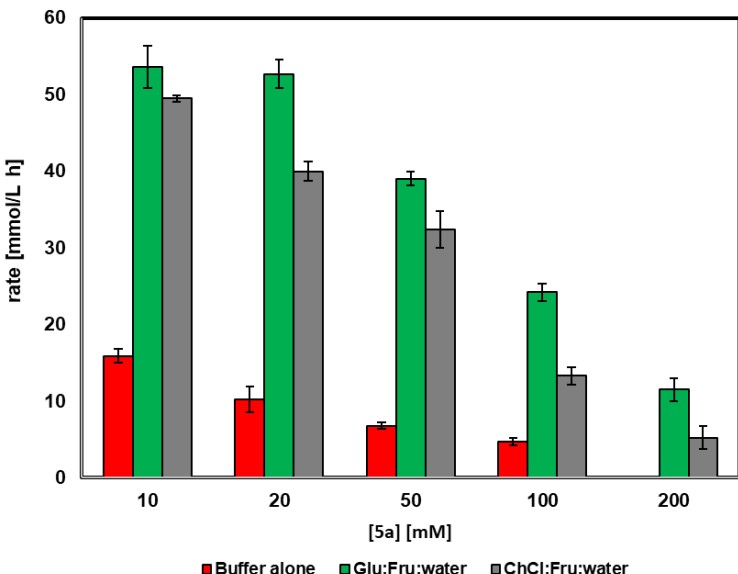

**Figure 1.** Effect of 4-chlorobenzyl alcohol concentration in the activity of HMFO in buffer or in buffer containing 60% *v/v* of the DES Glu:Fru:$H_2O$ (1:1:6) and ChCl:Fru:$H_2O$ (5:2:5).

As a further exploration of the use of DES as a cosolvent in HMFO-catalyzed reactions, the biooxidation of some alcohols was scaled up to multimilligram scale. Thus, 71.4 mg of 4-chlorobenzyl alcohol (**5a**) were dissolved in 25 mL of a mixture of Tris/HCl (50 mM, pH 8.0) and 60% *v/v* Glu:Fru:$H_2O$ (1:1:6), 1% *v/v* DMSO and HMFO (1.0 μM) at 30 °C for 24 h. A conversion of 88% was achieved. Product purification afforded aldehyde **5b** with an 80% yield. The reaction was also carried out in the same conditions in the absence of the DES. However, in that case, after 24 h of incubation, only 25% conversion was achieved, confirming the improvement obtained by the use of this DES in this process. The same procedure was followed in the biooxidation of (±)-1-indanol (**12a**). After 40 h (24% conversion), the reaction was stopped by extraction with EtOAc, which led to a 19% yield of 1-indanone (**12b**) after purification. The optical purity of the remaining 1-indanol was analyzed by HPLC. (*R*)-**12a** was obtained with a 25% *ee*, which corresponds to a moderate enantioselectivity value (*E* = 22). No ketone formation was observed when the oxidation was carried out in buffer.

To study the underlying effects on the catalytic performance of HMFO with different DESs, we tested how DESs affect (1) the solubility of molecular oxygen ($O_2$) and (2) the thermostability of the enzyme. Experiments regarding the $O_2$ solubility were performed using an oxygen-sensing instrument. This allowed determining the solubility of oxygen at atmospheric conditions and monitoring of oxygen depletion during conversion. As is shown in Table 4, it seems that the DESs have moderate effects on the oxygen solubility and therefore this could only influence the rate of catalysis to a small extent.

At least it is comforting to see that DESs support similar concentrations of dioxygen at atmospheric conditions. The DESs also were found to have specific effects on the rate of catalysis of HMFO with **5a**. This suggests that some of the DESs have an effect on the rate-limiting step in catalysis. Such an effect cannot be explained by a lower concentration of the electron acceptor, dioxygen, because that was not significantly different. Instead, the DES may promote a structural conformation that lowers the efficiency of substrate oxidation or product release. Among the different DESs, again Glu:Fru:H$_2$O (1:1:6) seems the most promising presenting the highest $k_{obs}$ for the reaction tested with DES. The thermostability of HMFO was also analyzed, by determining the apparent melting temperature ($T_m^{app}$) of the biocatalyst in the presence of mixtures containing buffer and 60% *v/v* of DES (Table 4) [45]. The presence of both DES (ChCl:Gly (1:2) and ChCl:Fru H$_2$O (5:2:5)) decrease the enzyme thermostability, as shown in Entries 2 and 3. This seems to correlate with lower activity in these DESs and is in line with DES-induced conformational changes. Interestingly, the natural DES, Glu:Fru:H$_2$O (1:1:6), has a strong stabilizing effect, as can be observed in Entry 4. At 60% *v/v* of this DES, the thermostability is even improved by 11.7 °C. This effect may explain the superior conversions observed in the reactions carried out with this DES. No melting temperature could be determined when testing ChCl:MA (1:1). As observed for the kinetic measurements (Table 4), HMFO does not seem to tolerate this specific DES.

**Table 4.** Effect of DESs (60% *v/v*) on solubility of dioxygen and activity of HMFO with alcohol **5a** [1].

| Entry | DES | O$_2$ (mM) | $k_{obs}$ (s$^{-1}$) | $T_m^{app}$ (°C) [2] |
|---|---|---|---|---|
| Entry 1 | None | 245 ± 0.4 | 9.7 ± 0.5 | 49.0 ± 0.0 |
| Entry 2 | ChCl:Gly (1:2) | 226 ± 4.9 | 2.4 ± 0.2 | 47.5 ± 0.35 |
| Entry 3 | ChCl:Fru H$_2$O (5:2:5) | 173 ± 9.2 | 3.3 ± 0.2 | 38.0 ± 0.7 |
| Entry 4 | Glu:Fru:H$_2$O (1:1:6) | 184 ± 2.8 | 7.1 ± 0.6 | 60.7 ± 0.35 |
| Entry 5 | ChCl:MA (1:1) | 222 ± 6.4 | 0.0 ± 0.0 | - |

[1] Experiments were performed using Oxigraph+; [2] Measured by ThermoFAD assay in Tris/HCl 50 mM pH 8.0 with/without DES.

To further probe the effect of the DES Glu:Fru:H$_2$O (1:1:6) on the performance of HMFO, different concentrations of this DES have been employed in the oxidation of HMF (**13a**). Conversion of HMF with HMFO typically leads to the formation of two main products: 5-formylfuran-2-carboxylic acid (FFA, **13b**) through double oxidation and FDCA (**13c**) through a triple oxidation (Scheme 3) [4]. When the reaction was performed in only buffer (Figure 2), complete oxidation of HMF was observed, with FFA as the major product (84%), whereas only 16% of the desired FDCA was obtained. The use of Glu:Fru:H$_2$O (1:1:6) up to 60% *v/v* had a significant and positive effect, as FDCA was obtained at a higher yield when compared with using buffer alone. An optimum of HMF to FDCA conversion was found at 30% DES, with 31% of the diacid **13c** formed after 24 h. This represents a substantial increase when compared with the oxidation performed in buffer. At 90% *v/v* DES, 13% of HMF is recovered after 24 h, whereas only 6% of FDCA is obtained. This demonstrates that a high concentration of this DES has a negative effect on the performance of HMFO. These results show that the use of this DES up to 60% *v/v* can improve the efficiency of oxidase-based biocatalytic processes.

**Scheme 3.** Oxidation of HMF **13a** to furan-2,5-dicarboxylic acid (FDCA) **13c** catalyzed by HMFO in the presence of the NADES Glu:Fru:H$_2$O (1:1:6).

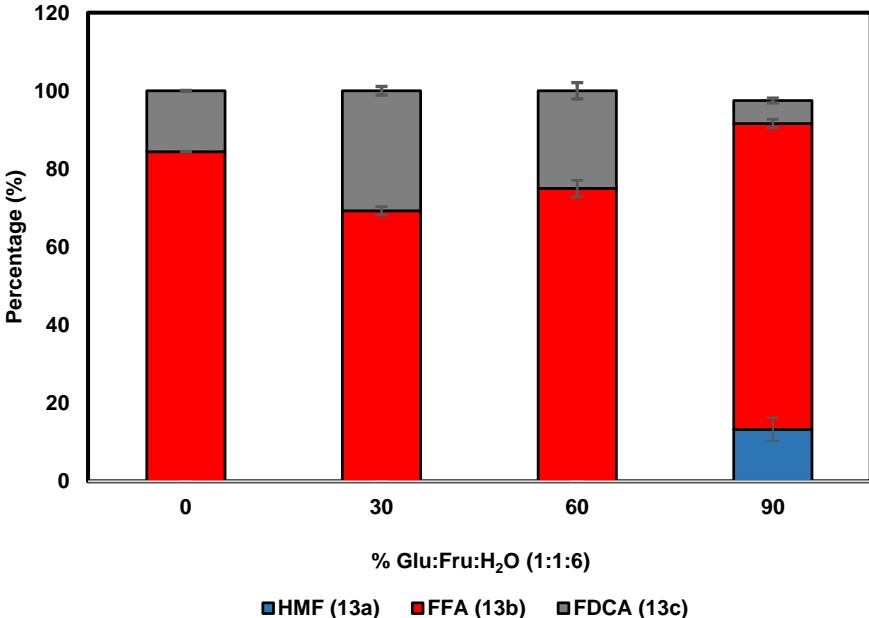

**Figure 2.** Effect of Glu:Fru:H$_2$O (1:1:6) concentration on the HMF (**13a**) oxidation catalyzed by HMFO.

## 3. Materials and Methods

### 3.1. Materials and Analytic Methods

Recombinant 5-hydroxymethylfurfural oxidase (HMFO) from *Methylovorus* sp. strain MP688 (UniProt accession number EAQP00) was overexpressed and purified according to previously described methods [2]. Catalase from *Micrococcus lysodeikticus* (150,000 units/mL) was obtained from Sigma-Aldrich. All other chemicals and analytical grade solvents were obtained from Acros Organics (Geel, Belgium), Sigma-Aldrich (Steinheim, Germany) and TCI Europe (Zwijndrecht, Belgium), and used without further purification.

Flash chromatography was performed using Merck silica gel 60 (230–400 mesh). $^1$H-NMR spectra were recorded with TMS (tetramethylsilane) as the internal standard, on a Bruker AC-300-DPX ($^1$H: 300.13 MHz) spectrometer. GC/MS analyses were performed with a GC Hewlett Packard 7890 Series II equipped with a Hewlett Packard 5973 chromatograph MS (Agilent Technologies, Santa Clara, USA) using an HP-5MS cross-linked methyl siloxane column (30 m × 0.25 mm x 0.25 μm, 1.0 bar N$_2$). To monitor levels of conversion, substrates and products were quantified by the use of calibration curves. HPLC analyses of alcohol **12a** were performed on a Thermo-Fisher (Waltham, USA) UltiMate chromatograph equipped with a Thermo UltiMate detector using a Chiralcel OD column (Daicel, Osaka, Japan, 0.46 cm × 25 cm). (*R*)-1-indanol configurations were established by comparing the HPLC chromatogram with the one described in the literature [46]. The products formed by HMFO using **13a** as a substrate were analyzed using an HPLC Jasco MD-2010 Plus (Oklahoma City, USA), equipped with a Zorbax Eclipse XDB-C8 column (Agilent, 0.46 cm × 15 cm, 5 μm).

### 3.2. General Procedure for the Enzymatic Oxidations Catalyzed by Isolated Wild Type HMFO

Unless otherwise stated, starting alcohols **1-13a** (10 mM) were dissolved in the corresponding mixtures 50 mM Tris/HCl buffer pH 8.0/DES at different concentrations (1.0 mL), containing 1% *v/v* dimethylsulfoxide (DMSO), catalase from *Micrococcus lysodeikticus* (10 μL, 150,000 U/mL) and HMFO (1.0 μM). Reactions were shaken at 220 rpm and 30 °C for the times indicated. Once finished, the crude reactions were extracted with EtOAc (3 × 500 μL). The organic phases were dried onto Na$_2$SO$_4$ and analyzed directly by GC/MS in order to determine the conversion of the biocatalyzed oxidations of alcohols **1-12a** (see Supplementary Material). The optical purity of alcohol **12a** was determined by

HPLC [46]. The conversion of the biooxidation of HMF (**13a**) was determined by HPLC, following the conditions described in the literature [2].

### 3.3. Biocatalyzed Oxidation of 4-Chlorobenzyl Alcohol at Multimilligram Scale

4-Chlorobenzyl alcohol (**5a**, 71.3 mg, 20 mM) was dissolved in a mixture of Tris/HCl 50 mM pH 8.0 (9.0 mL) and 13.5 mL of Glu:Fru:$H_2O$ (1:1:6) containing 0.25 mL of DMSO (1% *v/v*), catalase from *Micrococcus lysodeikticus* (250 μL, 150,000 U/mL) and HMFO (1.0 μM). The reaction was shaken during 45 h at 30 °C and 220 rpm. Once finished, $H_2O$ (10 mL), was added to the crude mixture, which was extracted with EtOAc (4 × 15 mL). The organic phases were dried onto $Na_2SO_4$ and the solvent was evaporated. The crude residue was analyzed by $^1$H-NMR (*c* = 89%) and purified by column chromatography using *n*-hexane:EtOAc 7:3 as eluent in order to obtain 56.2 mg of 4-chlorobenzaldehyde (80% yield).

### 3.4. Kinetic Resolution of Racemic 1-Indanol Catalyzed by Wild Type HMFO at Multimilligram Scale

1-Indanol (**12a**, 67 mg, 20 mM) was dissolved in a mixture of Tris/HCl 50 mM pH 8.0 (9.0 mL) and 13.5 mL of Glu:Fru:$H_2O$ (1:1:6) containing 250 μL of DMSO (1% *v/v*), catalase from *Micrococcus lysodeikticus* (250 μL, 150,000 U/mL) and HMFO (1.0 μM). The reaction was shaken during 45 h at 30 °C and 220 rpm. Once finished, $H_2O$ (10 mL), was added to the crude mixture, which was extracted with EtOAc (4 × 15 mL). The organic phases were dried onto $Na_2SO_4$ and the solvent was evaporated. The crude was analyzed by GC/MS (22% conversion) and purified by column chromatography using *n*-hexane/EtOAc 2:1 as eluent. After purification, 10.5 mg of 1-indanone (**12b**, 16% yield) and 49.7 mg of (*R*)-1-indanol (**12a**, 74% yield) were obtained. The optical purity of (*R*)-**12a** was measured by HPLC, which obtained a 25% *ee*.

### 3.5. Oxygen Depletion Analysis on Mixtures Buffer/DES

Reactions were performed using Oxigraph+ (Hansatech Instruments Ltd., Norfolk, England), with the following set-up: stirring 75, temperature 30 °C, atmospheric pressure and atmospheric dissolved $O_2$. Reactions were carried out in 500 μL of total reaction volume containing 60% of the corresponding DES, 50 mM Tris/HCl buffer pH 8.0 using wild type HMFO (0.5 μM) and 4-chlorobenzyl alcohol (**5a**, 1.25 mM). The $k_{obs}$ for each condition was calculated measuring the initial rate of the reaction.

### 3.6. Analysis of HMFO Stability in Mixtures Buffer/DES

Apparent melting temperature ($T_m^{app}$) was determined using the ThermoFAD assay [45]. Samples contained 60% of the corresponding DES, buffer Tris/HCl 50 mM pH 8.0 and wild type HMFO (20 μM). The $T_m^{app}$ were determined in duplo.

## 4. Conclusions

The present paper demonstrates the beneficial effects of DES on HMFO-catalyzed reactions. The use of NADES, composed of glucose and/or fructose, improved to a great extent the performance of HMFO from *Methylovorus* sp., a promising biocatalyst for the production of FDCA. This improvement may largely be due to the stabilizing effects of DESs on HMFO. The DES Glu:Fru:$H_2O$ (1:1:6) was shown to significantly increase its thermostability. HMFO was also found to be capable of converting a large number of alcohols into the corresponding aldehydes. Using 4-chlorobenzyl alcohol as the prototype substrate, it could be demonstrated that DES enables conversions at higher substrate concentrations. HMFO was also found to oxidize 1-indanol to 1-indanone in a moderately enantioselective manner. The use of the NADES Glu:Fru:$H_2O$ (1:1:6) was finally applied in the optimization of the oxidation of HMF to FDCA. This revealed that at 30% *v/v* DES, a relatively high amount of FDCA could be obtained. These findings may be important for further optimization of HMFO- other oxidase-based biotechnological processes.

**Supplementary Materials:** The following are available online at http://www.mdpi.com/2073-4344/10/4/447/s1, Table S1: Retention times at GC/MS analyses in the biooxidations of alcohols **1-12a**, Table S2: Percentage of products formed during the oxidation of HMF catalyzed by HMFO in Tris/HCl 50 mM pH 8.0 containing Glu:Fru:$H_2O$ (1:1:6).

**Author Contributions:** G.d.G. and C.M. performed the experiments and G.d.G., C.M. and M.W.F. conceived, designed and wrote the paper. All authors have read and agreed to the published version of the manuscript.

**Funding:** G.d.G. thanks MINECO for Ramón y Cajal Program for personal funding.

**Conflicts of Interest:** The authors declare no conflicts of interest.

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
