# Peer review of "Positive Impact of Natural Deep Eutectic Solvents on the Biocatalytic Performance of 5-Hydroxymethyl-Furfural Oxidase"

_catalysts, doi:10.3390/catal10040447_

Round 1

Reviewer 1 Report

The manuscript "Positive Impact of Natural Deep Eutectic Solvents on the Biocatalytic Performance of 5-Hydroxymethyl Furfural Oxidase" (Manuscript ID: catalysts-762263) is devoted to the study of the 5-hydroxymethylfurfural oxidase-catalysed oxidation of alcohols in a natural deep eutectic solvent medium. The studied reactions are not new, but they proceed under new conditions.

My comments concerning the manuscript are given below.

Title, row 16, etc.: The space between "5-hydroxymethyl" and "furfural" should be deleted.

Row 17: It is better to replace the phrase "other oxidized aromatic compounds" to "other compounds, such as carbonyl compounds and carboxylic acids".

Row 67: The terminus "aromatic alcohol" means phenol. Therefore the word "aromatic" must be deleted.

Rows 80-81 and Scheme 1: There are inconsistencies between the compound labels in the text and in Scheme 1.

Scheme 1: The general structures of the starting material and product (RCOH) should be corrected, because not only primary alcohols are involved in this reaction.

Product yields should be given in addition to the conversion of starting materials.

Row 254: Which calibration curves were used in the GC analysis? Which internal standard was used, if the internal standard method was applied? 

Row 282: It is impossible to prepare 49.7 mg of the R-isomer from 67 mg of the racemate by separation. As I understand, the mixture enriched in the R-isomer was obtained. The sentence "After purification, 10.5 mg of 1-indanone (12b, 16% yield) and 49.7 mg of (R)-1-indanol (12a, 74% yield) were obtained." should be corrected.

I think, the manuscript can be published in Catalysts after a major revision.

Author Response

The manuscript "Positive Impact of Natural Deep Eutectic Solvents on the Biocatalytic Performance of 5-Hydroxymethyl Furfural Oxidase" (Manuscript ID: catalysts-762263) is devoted to the study of the 5-hydroxymethylfurfural oxidase-catalysed oxidation of alcohols in a natural deep eutectic solvent medium. The studied reactions are not new, but they proceed under new conditions.

My comments concerning the manuscript are given below.

Title, row 16, etc.: The space between "5-hydroxymethyl" and "furfural" should be deleted.

This has been corrected

Row 17: It is better to replace the phrase "other oxidized aromatic compounds" to "other compounds, such as carbonyl compounds and carboxylic acids".

We have replaced this sentence as suggested by the referee

Row 67: The terminus "aromatic alcohol" means phenol. Therefore, the word "aromatic" must be deleted.

Aromatic alcohols seem to be defined differently, depending on the text book or source. For clarity, we have now explained what is meant with aromatic alcohols (alcohols with an aromatic moiety) in line 67

Rows 80-81 and Scheme 1: There are inconsistencies between the compound labels in the text and in Scheme 1.

We have corrected Scheme 1 by drawing correctly the structures of compounds 8a to 11a, that were wrongly placed.

Scheme 1: The general structures of the starting material and product (RCOH) should be corrected, because not only primary alcohols are involved in this reaction.

We have corrected the general structures of both substrate and product in Scheme 1 by indicating R1 and R2 as the substituents.

Product yields should be given in addition to the conversion of starting materials.

The reactions to determine the substrate profile and the effect of the (NA)DES in HMFO have been carried out at 1.0 mg scale, so only measurements of conversion have been performed. For this reason, we have performed additional experiments at multimilligram scale, in order to determine the yield under these conditions.

Row 254: Which calibration curves were used in the GC analysis? Which internal standard was used, if the internal standard method was applied? 

Conversions were determined by GC/MS by performing a calibration of the substrates and the products at different concentrations, in order to obtain the response of each compound to the detector and then make the calculations with these values. This has been mentioned now in the Electronic Supporting Information, at Section 1, focused on the GC/MS analyses.  

Row 282: It is impossible to prepare 49.7 mg of the R-isomer from 67 mg of the racemate by separation. As I understand, the mixture enriched in the R-isomer was obtained. The sentence "After purification, 10.5 mg of 1-indanone (12b, 16% yield) and 49.7 mg of (R)-1-indanol (12a, 74% yield) were obtained." should be corrected.

We have obtained this yield of the final product because the reaction proceeded with a conversion of only 24%. This means that only 24% of the ketone was obtained, whereas 76% of the alcohol was recovered. After the purification, the yield of alcohol was 74% and the yield of ketone 16%. This is the reason of the high amount of starting material recovered in the kinetic resolution, the low conversion of the reaction after 40 hours.

Reviewer 2 Report

In the present manuscript the authors report the effect of deep eutectic solvents (DES) on the oxidation of various alcohols by 5-Hydroxymethyl - Furfural Oxidase. The results presented are quite interesting, given that, with the use of DES, an increase of the conversion yield was observed for various substrates. However, the manuscript is not suitable for publication in its present form, due to two significant omissions: First of all, the authors do not report the standard deviations of any of their results presented in this manuscript, therefore, their results are not reliable at present. The results should be derived from at least two independent experiments, and the errors must be provided.  Second of all, the discussion section is missing from the manuscript. The authors simply present their results without any comparison with the relative literature, and without discussing their findings in any context.

Other issues are as follows:

L28: The EC number for HMFO should be included, and preferably the UniProt accession number should be included in the Materials and Methods section.

Scheme 1: 12a compound is missing.

Lines 177-180: In the large scale reaction, DMSO was not used?

Lines 254-259: Please provide in detail the HPLC protocols used for the quantification of the conversion of substrates 12a and 13a, including the solvents used.

Line 292: Please provide a suitable reference for the ThermoFAD assay.

Line 298-299: Unfortunately this conclusion is not supported by the results, since the use of most of the tested DES resulted in lower thermostability. The only DES enhancing the thermostability of HMFO was the DES Glu:Fru:H2O (1:1:6), but, this results could be due to an experimental error, since standard deviations are not provided.

Lines 305-306: ‘This revealed that at 30% v/v DES, a relatively high amount of FDCA could be obtained’ The use of the word ‘relatively’ should be omitted, since no literature is provided for comparison.

Author Response

In the present manuscript the authors report the effect of deep eutectic solvents (DES) on the oxidation of various alcohols by 5-Hydroxymethyl - Furfural Oxidase. The results presented are quite interesting, given that, with the use of DES, an increase of the conversion yield was observed for various substrates. However, the manuscript is not suitable for publication in its present form, due to two significant omissions: First of all, the authors do not report the standard deviations of any of their results presented in this manuscript, therefore, their results are not reliable at present. The results should be derived from at least two independent experiments, and the errors must be provided.  Second of all, the discussion section is missing from the manuscript. The authors simply present their results without any comparison with the relative literature, and without discussing their findings in any context.

Regarding the referee first issue, all the oxidation herein shown have been performed at least in duple. We have now included the standard deviations of all the experiments performed.

The main objective of this manuscript was to show the positive impact of a natural deep eutectic solvent on the performance of HMFO. As for other studies, rationalizing the positive or negative effect of a solvent in a biocatalyst is really complicated. In most medium engineering biocatalytic approaches there are no exact explanations of the effect of a specific solvent. In most cases, the solvent can affect structural properties of the enzyme by inducing (local) conformational changes, dissociation of oligomers or even partial unfolding. Yet, it can also have very specific effects by binding in or near the active site, affecting the affinity, efficiency and/or selectivity of the enzyme. And except for having an effect on the biocatalyst, solvents influence medium properties, resulting in for example, improved substrate solubility. The effects can also be mixtures of any of these aforementioned effects. Though our study does not give a precise answer to why a specific NADES is beneficial, we have determined and excluded the effect of NADES on availability/solubility of dioxygen. This is an important parameter for oxygen-utilizing enzymes such as oxidases and monooxygenases and had, to our knowledge, not been reported before. We have shown that NADES do not hamper the use of dioxygen. We have added some more references as part of the discussion to put this study in the context of some other recent studies that demonstrated the use of (NA)DES in biocatalysis. Nevertheless, for clarity,   we have included a paragraph on the results section in which the use of NADES is shown to be beneficial for redox biocatalysts, stating that this is the first demonstration for using a flavin-dependent oxidase in presence of (NA)DES.

Other issues are as follows:

L28: The EC number for HMFO should be included, and preferably the UniProt accession number should be included in the Materials and Methods section.

We have included the HMFO EC number (1.1.3.47) at row 28, the first time it is mentioned in the Introduction. The UniProt accession number has been placed in the Materials and Methods section.

Scheme 1: 12a compound is missing.

Scheme 1 has been corrected, including all the compounds tested.

Lines 177-180: In the large scale reaction, DMSO was not used?

DMSO is employed in the large scale conversion, as indicated at the Materials and Methods Section (Line 278). For clarity in the Results and Discussion section, we have included it at Line 179

Lines 254-259: Please provide in detail the HPLC protocols used for the quantification of the conversion of substrates 12a and 13a, including the solvents used.

The determination of the conversion inthe biooxidation of 1-indanol (12a) to 1-indanone (12b) was performed by GC/MS. HPLC was employed to determine the optical purity of the optically active 1-indanol. Regarding the HPLC protocol for the biooxidation of 13a (HMF), it has been included in the original version at the Electronic Supporting Information (together with all the chromatographic analyses), but in addition, we have included a reference at the Materials and Method section in order to provide it.

Line 292: Please provide a suitable reference for the ThermoFAD assay.

This reference has been provided in the original paper and corresponds to reference 43: Forneris F, Orru R, Bonivento D, Chiarelli L R, Mattevi A. ThermoFAD, a Thermofluor-adapted flavin ad hoc detection system for protein folding and ligand binding. FEBS J. 2009, 276, 2833-2840.

Line 298-299: Unfortunately this conclusion is not supported by the results, since the use of most of the tested DES resulted in lower thermostability. The only DES enhancing the thermostability of HMFO was the DES Glu:Fru:H2O (1:1:6), but, this results could be due to an experimental error, since standard deviations are not provided.

The determined melting temperatures are very precise but we forgot to include the standard deviation. We have added this information (s.d. < 0.4 °C) to the materials and methods. Thus, the indicated difference in melting temperatures (<11°C) is significant and worth noting. 

Lines 305-306: ‘This revealed that at 30% v/v DES, a relatively high amount of FDCA could be obtained’ The use of the word ‘relatively’ should be omitted, since no literature is provided for comparison.

This has been corrected as pointed by the referee.

Reviewer 3 Report

This is an interesting study that reports on the use of an oxidase enzyme for conversion of alcohols and aldehydes.  The use of the novel solvent systems in many cases have a positive effect on yield.

Major comments:

  1. The term “deep eutectic solvent” is sometimes used specifically to refer to a mixture of a quaternary ammonium compound and a hydrogen bond donor, though here the term also encompasses solvent systems composed of two sugars and water. It would be helpful if the Authors explained in the introduction their definition of deep eutectic solvents and the extent to which the different solvent systems employed exhibit eutectic properties. The Authors also need to explain whether the various solvent systems mixed to produce a single liquid phase or whether multiple phases were observed.
  2. The oxidase reactions employed are expected to generate a stoichiometric amount of hydrogen peroxide. It would be helpful to state if any problems are anticipated due to this, whether a system for removing it (e.g. catalase) should be employed when the systems are used and whether it is possible that the hydrogen peroxide reacts with the components of the solvent.
  3. No replicates or tests of statistical significance of the results are presented. The study would be greatly strengthened by information about the reproducibility of the results.

Minor comments:

  1. Line 14.  “the last years” would be better as “recent years”.  “its use” should be “their use”.
  2. Line 35. “it was possible” should be “was it possible”.
  3. Lines 79-82. It needs to be stated in the text that the (negative) results for compounds 8a, 9a, 10a and 11a are not shown in Table 1.
  4. Scheme 1 is misleading in several minor details. It shows the reactions conducted only in the buffer/DES/DMSO systems, when in fact they were also conducted in aqueous buffer only.  The reaction scheme shows compounds 1a-12a, when in fact only 11 structures are shown below.  The structure shown for 11a is 1-indanol,  whereas in the text (line 82) it is stated that this compound is 1-phenylethanol (12a should be 1-indanol).  Lastly, the reaction paradigm at the top is specific to primarly alcohols, whereas two of the substrates (1-phenylethanol and 1-indanol) are secondary alcohols.
  5. Line 99. “both” here suggests that a mixture of the two solvents ChCl:Gly and ChCl:EG was used, which I believe is not what the Authors mean.  I suggested deleting “both” and replacing “and” with “or”.
  6. Line 131 and elsewhere. It is helpful that the entries in the tables are numbered, though potentially confusing that the same entry number refers to different reactions in different tables.  Please be careful to repeat the table numbers sufficiently often that the reader is always clear exactly which reaction is being referred to.
  7. Line 137-139. I found the sentence beginning “The optical purity of the starting alcohol…” very confusing.  This sentence needs rewriting to be clear about what the Authors mean.  It was not clear to me whether the ee of the R-alcohol was 17% before the start of the reaction or whether it started at zero and increased to 17 % as the reaction progressed and the S-alcohol was preferentially oxidised.
  8. Line 140. “one of the first examples of enantioselective oxidation catalysed by HFMO” – if there are previous examples in the literature, these need to be referred to.
  9. Line 149. “buffer” would be better as “buffer alone”.  Similarly, in the key to the coloured bars in Figure 1, the label “buffer” should read “buffer alone” to avoid the implication that the other reactions do not contain buffer.
  10. Lines 303-304. “in an enantioselective manner” may be better as “in a moderately enantioselective manner”.

Author Response

This is an interesting study that reports on the use of an oxidase enzyme for conversion of alcohols and aldehydes.  The use of the novel solvent systems in many cases have a positive effect on yield.

Major comments:

  1. The term “deep eutectic solvent” is sometimes used specifically to refer to a mixture of a quaternary ammonium compound and a hydrogen bond donor, though here the term also encompasses solvent systems composed of two sugars and water. It would be helpful if the Authors explained in the introduction their definition of deep eutectic solvents and the extent to which the different solvent systems employed exhibit eutectic properties. The Authors also need to explain whether the various solvent systems mixed to produce a single liquid phase or whether multiple phases were observed.

We have added a sentence to explain that the mixtures did not show any phase separation.

Regarding the composition of NADES, sugars mixtures including sucrose, fructose or glucose can take part of them and there are some examples in which the sugar mixtures are NADES. For instance, at González et al. Flavour Frag. J. 2018, 33, 91-96, different sugar mixtures, including Glu:Fru:H2O (1:1:6), are employed as NADES for vanillin extraction. Also at Choi et al. Plant Physiology, 2011, 156, 1701-1706, a glucose-fructose DES is employed.

  1. The oxidase reactions employed are expected to generate a stoichiometric amount of hydrogen peroxide. It would be helpful to state if any problems are anticipated due to this, whether a system for removing it (e.g. catalase) should be employed when the systems are used and whether it is possible that the hydrogen peroxide reacts with the components of the solvent.

Catalase has been employed in all the oxidations tested in order to remove the hydrogen peroxide formed during the oxidations. We apologize for this omission: it has now been indicated in the manuscript.

  1. No replicates or tests of statistical significance of the results are presented. The study would be greatly strengthened by information about the reproducibility of the results.

As also indicated to referee 2, all the experiments have been performed at least twice. We had only included the medium value, but not the standard deviations. In order to improve the quality of the manuscript, we have included the standard deviation values in this revised version.

Minor comments:

  1. Line 14.  “the last years” would be better as “recent years”.  “its use” should be “their use”.

Line 14 has been corrected as suggested by the referee.

  1. Line 35. “it was possible” should be “was it possible”.

We have modified this as suggested.

  1. Lines 79-82. It needs to be stated in the text that the (negative) results for compounds 8a, 9a, 10a and 11a are not shown in Table 1.

We have included a sentence in brackets in which we state that the results for compounds 8-11a are not shown in Table 1.

  1. Scheme 1 is misleading in several minor details. It shows the reactions conducted only in the buffer/DES/DMSO systems, when in fact they were also conducted in aqueous buffer only.  The reaction scheme shows compounds 1a-12a, when in fact only 11 structures are shown below.  The structure shown for 11a is 1-indanol,  whereas in the text (line 82) it is stated that this compound is 1-phenylethanol (12a should be 1-indanol).  Lastly, the reaction paradigm at the top is specific to primarly alcohols, whereas two of the substrates (1-phenylethanol and 1-indanol) are secondary alcohols.

Scheme 1 has been modified in order to include 1-phenylethanol with its right number, and the structure of substrates and products has been modified to include secondary alcohols as well as primary alcohols.

  1. Line 99. “both” here suggests that a mixture of the two solvents ChCl:Gly and ChCl:EG was used, which I believe is not what the Authors mean.  I suggested deleting “both” and replacing “and” with “or”.

We have modified this sentence as indicated by the referee in order to make it clearer.

  1. Line 131 and elsewhere. It is helpful that the entries in the tables are numbered, though potentially confusing that the same entry number refers to different reactions in different tables.  Please be careful to repeat the table numbers sufficiently often that the reader is always clear exactly which reaction is being referred to.

We have included the Table numbers in the text of the manuscript in order clarify possible misunderstandings, as shown by the referee for instance at line 131.

  1. Line 137-139. I found the sentence beginning “The optical purity of the starting alcohol…” very confusing.  This sentence needs rewriting to be clear about what the Authors mean.  It was not clear to me whether the ee of the R-alcohol was 17% before the start of the reaction or whether it started at zero and increased to 17 % as the reaction progressed and the S-alcohol was preferentially oxidised.

Herein we wanted to refer to the optical purity of 1-indanol after finishing the reaction. It is true that the term starting alcohol can be confusing, so we have deleted it and indicate “The optical purity of 1-indanol at the end of the reaction….”.

  1. Line 140. “one of the first examples of enantioselective oxidation catalysed by HFMO” – if there are previous examples in the literature, these need to be referred to.

In this sentence, we were referring to reference 12, the previous example of enantioselective oxidation catalysed by HMFO. For this reason, we have included it at the end of the sentence.

  1. Line 149. “buffer” would be better as “buffer alone”.  Similarly, in the key to the coloured bars in Figure 1, the label “buffer” should read “buffer alone” to avoid the implication that the other reactions do not contain buffer.

We have corrected this in the text (line 149) and at Figure 1.

  1. Lines 303-304. “in an enantioselective manner” may be better as “in a moderately enantioselective manner”.

We have included “in a moderately enantioselective manner” in Lines 303-304.

Round 2

Reviewer 1 Report

Now I recommend the manuscript "Positive Impact of Natural Deep Eutectic Solvents on the Biocatalytic Performance of 5-Hydroxymethylfurfural Oxidase" for publication in Catalysts.

Author Response

We agree with this referee. We have made some modifications in the English of the manuscript.

Reviewer 2 Report

The authors of the present manuscript performed most of the suggested revisions, but unfortunately the manuscript still is not fit for publication. First of all, although they introduced the standard deviations of their data, error bars are still nowhere to be found in the graphs, raising concerns about the credibility of the study. Second of all, in the revised version, there is still not a discussion section, which is not suitable for a scientific paper. The authors added a few references in the conclusion section, which is not correct, the conclusion is not the place to reference other studies. Also, in my previous report I suggested to rephrase their conclusions, since no comparison in the literature was made, and this change has not been performed. However, the authors did try to discuss their findings in more detail, but unfortunately the added paragraph (lines 94-103 in the revised version) still does not constitute a proper discussion of their findings in a broad context. I would suggest to add the error bars to the diagrams, and write a more suitable discussion, which should also be included in the section title (Results and Discussion). 

Author Response

"The authors of the present manuscript performed most of the suggested revisions, but unfortunately the manuscript still is not fit for publication.”

Indeed, we had responded to the suggestions of the reviewers. It is disappointing to read that this was not sufficient for the reviewer. We have used the comments below to improve the manuscript.

“First of all, although they introduced the standard deviations of their data, error bars are still nowhere to be found in the graphs, raising concerns about the credibility of the study.”

We have added information on the accuracy of the data presented in the two figures of the manuscript: Figure 1 and Figure 2 have been replotted with error bars. All table display standard errors. We are disappointed that the reviewer questions the credibility of our work. We have described all experiments in detail and all materials and methods. All experiments can be reproduced using this information. Also, all experiments have been logged in labjournals. There is no reason to doubt the credibility: this is a very harsh and unnecessary statement.

“Second of all, in the revised version, there is still not a discussion section, which is not suitable for a scientific paper. The authors added a few references in the conclusion section, which is not correct, the conclusion is not the place to reference other studies. Also, in my previous report I suggested to rephrase their conclusions, since no comparison in the literature was made, and this change has not been performed. However, the authors did try to discuss their findings in more detail, but unfortunately the added paragraph (lines 94-103 in the revised version) still does not constitute a proper discussion of their findings in a broad context. I would suggest to add the error bars to the diagrams, and write a more suitable discussion, which should also be included in the section title (Results and Discussion)."

The reviewer insists on having more discussion on the results, and to make clear that results have been discussed. We have now changed the caption of the results section into “Results and Discussion” as suggested. We also added some more lines of discussion in the respective section. We refrain from a very extensive discussion as that would be very speculative and distract from the obtained results and insights. With the current presentation of results, discussion and 47 provided references. We feel that this is quite sufficient.

As the reviewer does not appreciate the use of references in the Conclusion section, we have omitted them as they were not essential. We don’t understand the contradictory statements concerning the Conclusion section that (1) “the conclusion is not the place to reference other studies”, and (2) the suggestion “to rephrase their conclusions, since no comparison in the literature was made”. We feel that the Conclusion gives a good overview of the obtained results and insights. We realize that everyone would discus results in a different matter and write conclusions differently, but with having presented all results in a logic manner, the reader can make conclusions that are relevant for him or her.

With best regards

Gonzalo de Gonzalo and Marco W. Fraaije